# Environmental and Lifestyle Cancer Risk Factors: Shaping Extracellular Vesicle OncomiRs and Paving the Path to Cancer Development

**DOI:** 10.3390/cancers15174317

**Published:** 2023-08-29

**Authors:** Valentina Bollati, Paola Monti, Davide Biganzoli, Giuseppe Marano, Chiara Favero, Simona Iodice, Luca Ferrari, Laura Dioni, Francesca Bianchi, Angela Cecilia Pesatori, Elia Mario Biganzoli

**Affiliations:** 1Epiget Lab, Department of Clinical Sciences and Community Health, University of Milan, 20133 Milan, Italy; paola.monti@unimi.it (P.M.); chiara.favero@unimi.it (C.F.); simona.iodice@unimi.it (S.I.); luca.ferrari@unimi.it (L.F.); laura.dioni@unimi.it (L.D.); angela.pesatori@unimi.it (A.C.P.); 2Occupational Health Unit, Fondazione IRCCS Ca’ Granda Ospedale Maggiore Policlinico, 20122 Milan, Italy; 3Center of Functional Genomics and Rare Diseases, Buzzi Children’s Hospital, 20154 Milan, Italy; davide.biganzoli@asst-fbf-sacco.it; 4Unit of Medical Statistics, Bioinformatics and Epidemiology, Department of Biomedical and Clinical Sciences (DIBIC), University of Milan, 20133 Milan, Italy; giuseppe.marano@unimi.it; 5Dipartimento di Scienze Biomediche per la Salute, University of Milan, 20133 Milan, Italy; francesca.bianchi@unimi.it; 6U. O. Laboratorio Morfologia Umana Applicata, IRCCS Policlinico San Donato, 20097 Milan, Italy

**Keywords:** oncomiR, exposome, lifestyle factors, environmental exposures, SPHERE population, MARS models

## Abstract

**Simple Summary:**

In this study, we aim to shed light on a fascinating new aspect of cellular communication involving extracellular vesicles (EVs) and their cargo of microRNAs (EV-miRNAs). These EVs were once thought of as cellular waste but are now known to play a crucial role in transferring vital information between cells. We investigated how EV-miRNAs respond to various environmental and lifestyle factors and how this might be linked to the development and progression of cancer. By analyzing the associations between these factors and specific miRNAs related to common cancers (oncomiRs), we aim to better understand the complex regulatory networks of miRNAs in response to exogenous influences. The findings could revolutionize our understanding of cancer development and have important implications for future exposome studies in the research community.

**Abstract:**

Intercellular communication has been transformed by the discovery of extracellular vesicles (EVs) and their cargo, including microRNAs (miRNAs), which play crucial roles in intercellular signaling. These EVs were previously disregarded as cellular debris but are now recognized as vital mediators of biological information transfer between cells. Furthermore, they respond not only to internal stimuli but also to environmental and lifestyle factors. Identifying EV-borne oncomiRs, a subset of miRNAs implicated in cancer development, could revolutionize our understanding of how environmental and lifestyle exposures contribute to oncogenesis. To investigate this, we studied the plasma levels of EV-borne oncomiRs in a population of 673 women and 238 men with a body mass index > 25 kg/m^2^ (SPHERE population). The top fifty oncomiRs associated with the three most common cancers in women (breast, colorectal, and lung carcinomas) and men (lung, prostate, and colorectal carcinomas) were selected from the OncomiR database. Only oncomiRs expressed in more than 20% of the population were considered for statistical analysis. Using a Multivariate Adaptive Regression Splines (MARS) model, we explored the interactions between environmental/lifestyle exposures and EV oncomiRs to develop optimized predictor combinations for each EV oncomiR. This innovative approach allowed us to better understand miRNA regulation in response to multiple environmental and lifestyle influences. By uncovering non-linear relationships among variables, we gained valuable insights into the complexity of miRNA regulatory networks. Ultimately, this research paves the way for comprehensive exposome studies in the future.

## 1. Introduction

Environmental and lifestyle factors exert a profound influence on chronic diseases, particularly in the context of cancer development and related clinical outcomes [1,2]. These factors encompass diverse elements [3,4], ranging from exposure to environmental carcinogens and dietary habits [5] to physical activity levels [6], adiposity [7], smoking, alcohol consumption [8], and socioeconomic status [9]. The intricate interplay between these factors and cancer etiology underscores the necessity for a comprehensive approach to cancer prevention and management. By addressing the environmental and lifestyle determinants of cancer, we can implement targeted interventions and promote healthier behaviors [10], aiming to reduce non-intrinsic cancer risks and enhance clinical outcomes in a personalized way.

Individuals with pre-existing hypersusceptibility conditions, such as those associated with adiposity (excessive body fat), might experience an amplified influence from the exposome, i.e., an accumulation of lifelong environmental exposures [11,12]. Adiposity-related conditions, like being overweight/obesity and metabolic syndrome, foster a hormonally dysregulated and pro-inflammatory environment both at systemic and tissue levels. This inflamed host environment can synergize with external exposures, augmenting the vulnerability to cancer development and progression [11,13].

Extracellular vesicles (EVs), encompassing exosomes and microvesicles, are small membrane-bound structures released by various cell types [14]. Once considered cellular debris, EVs are now recognized as sophisticated carriers of cellular cargo, including proteins, lipids, and nucleic acids [15]. Owing to their ability to traverse biological barriers, EVs serve as conduits for intercellular communication and facilitate information exchanges between cells, even across distant locations within the body [14]. These distinctive attributes render EVs versatile entities with significant roles in diverse physiological processes and disease contexts, including cancer [16,17].

MicroRNAs (miRNAs) are pivotal regulators of gene expression among the myriad of molecules transported by EVs. MiRNAs, short non-coding RNA molecules, play a critical role in post-transcriptional gene regulation by targeting messenger RNAs (mRNAs) for degradation or translational inhibition [18]. Within this group, oncomiRs, a subset of miRNAs, have garnered attention for their involvement in cancer-related processes, including tumor initiation, growth, metastasis, and therapy resistance [19,20].

Understanding the mechanisms driving the selective packaging of oncomiRs into EVs and their subsequent release into the extracellular space has become central in cancer research. EVs enriched with oncomiRs can engage in cell-to-cell communication, delivering functional oncomiRs to recipient cells—ranging from neighboring cancer cells to fibroblasts, adipocytes, and immune cells. This transfer reshapes the gene expression profiles and behavior of recipient cells, contributing to the ability of cancer cells to shape both local and distant microenvironments. EVs containing miRNAs thus dynamically orchestrate multidirectional communication, serving as the hubs of the intercellular network, ultimately transforming the microenvironment in a pro-tumoral manner. This intriguing phenomenon fuels interest in cancer biology as it may hold substantial implications for innovative diagnostics and therapeutic strategies.

The SPHERE project [21], with its unique attributes, has enabled the investigation of EVs and their miRNA content’s role in mediating the impact of air particulate matter (PM) on cardiovascular health. The SPHERE study offers comprehensive insights into the microRNA content of EVs within a population of over 2000 individuals characterized by a body mass index (BMI) exceeding 25 and a well-defined exposome.

In this paper, we delve into the captivating domain of EVs and oncomiRs, probing the oncomiRs associated with specific cancer types based on their dependency on major exposome factors in the SPHERE screening subpopulation of 911 non-oncological subjects with assessed miRNome.

Given the complexity of analyzing these data, it is essential to consider the application of inductive inference methods rooted in machine learning, extending multivariate analysis and regression approaches. In this context, adopting automated methods for variable selection and their preferably non-linear/non-additive effects on EV-contained miRNA expression is pivotal. Within the realm of multiple regression models, the Multivariate Adaptive Regression Splines (MARS) methodology appears promising, given its ability to automatically identify non-linear effects and interactions (effect modifiers) among predictors.

## 2. Materials and Methods

### 2.1. Study Design and Participants

We recruited 911 subjects who were overweight/obese from the Center for Obesity and Work at IRCCS Fondazione Ca’Granda—Ospedale Maggiore Policlinico, Milan, Italy, between September 2010 and March 2015 (recruitment of the subset reported in the present paper ended on January 2013). This was the screening subset of the cross-sectional study, SPHERE (Susceptibility to Particle Health Effects, miRNAs, and Exosomes), funded by the European Research Council (ERC); it focused on investigating the interplay between particle health effects, miRNAs, and EVs [21,22]. Briefly, the eligibility criteria included being over 18 years old, having a BMI ≥ 25 kg/m^2^, residing in the Lombardy region (Italy), and providing informed consent and blood/urine samples. Exclusions were recent (i.e., in the last 5 years) cancer, heart disease, stroke, or other chronic conditions. This study followed the Helsinki Declaration principles and received ethics committee approval from the Fondazione IRCCS Ca’Granda Ospedale Maggiore Policlinico di Milano (approval number 1425). The participation rate was 92%.

### 2.2. Isolation and Purification of EVs and miRNA-EVs from Plasma

EV preparation and miRNA analysis have been described elsewhere [22]. Briefly, blood was collected in EDTA tubes in the morning and transported to the EPIGET Lab (University of Milan) within 2 h. Blood was centrifuged at 1200× *g* for 15 min at room temperature to obtain plasma, followed by additional centrifugation steps (1000, 2000, and 3000× *g* for 15 min at 4 °C) to remove debris. Plasma was ultracentrifuged (Beckman Coulter Optima-MAX-XP, Brea, CA, USA) at 110,000× *g* for 75 min at 4 °C to obtain an EV-rich pellet, which was resuspended in filtered PBS. MiRNAs were isolated from frozen EV pellets using the miRNeasy Kit and RNeasy CleanUp Kit (Qiagen, Germantown, MD, USA) and stored at −80 °C.

### 2.3. miRNA Analysis

For reverse transcription (RT), Megaplex™ RT Primers (Pool A v2.1 and Pool B v3.0) and the TaqMan^®^ Micro RNA Reverse Transcriptase Kit (Life Technologies, Foster City, CA, USA) were used. Two reactions were performed to cover 754 target miRNAs, including 16 replicates of 4 internal controls (ath-miR159a, RNU48, RNU44, and U6). Each reaction consisted of 0.75 μL of Megaplex RT Primers (Pool A or Pool B), 0.15 μL of dNTPs, 0.75 μL of 10× RT Buffer, 0.90 μL of MgCl2, 0.1 μL of RNase Inhibitor, 1.5 μL of MultiScribe™ Reverse Transcriptase, and 3.3 μL of miRNAs. After incubation on ice for 5 min, the mixture underwent a thermal protocol in a C1000 Thermal Cycler (Biorad, Hercules, CA, USA) with specific temperature cycles. The resulting cDNA samples were stored at −20 °C until further use. For preamplification, each cDNA requiring preamplification was loaded onto a 96-well plate and combined with TaqMan^®^ PreAmp Master Mix, nuclease-free water, and Megaplex™ PreAmp Primers. The preamplification reaction involved specific thermal conditions and the preamplified samples were stored at 4 °C until analysis with the OpenArray^®^ System. The preamplified cDNA was diluted and mixed with TaqMan OpenArray^®^ Real-Time PCR Master Mix. The reaction mix was aliquoted into the wells of a 384-well OpenArray^®^ plate using the MicroLab STAR Let instrument (Hamilton Robotics, Birmingham, UK). The plate was then loaded into a TaqMan™ OpenArray^®^ Human miRNA Panel using the QuantStudio™ AccuFill System Robot (Life Technologies, Foster City, CA, USA). Finally, the mixture was analyzed with the QuantStudio™ 12K Flex Real-Time PCR System and the OpenArray^®^ Platform (QS12KFlex) following the manufacturer’s instructions.

In total, 545 miRNAs were included in the analysis after the exclusion of non-amplified miRNAs. The application of the global mean was identified as the optimal normalization method; miRNA expression levels were quantified using relative quantification, represented as 2^−ΔCt^. Leveraging the OncomiR database [23], we extracted the top 50 oncomiRs for each of the 3 most prevalent cancers in both women (breast, colorectal, and lung carcinomas) and men (lung, prostate, and colorectal carcinomas) [24]. Notably, this selection process was underpinned by stringent criteria, including a significance threshold of a *p*-value < 0.0001 and a false discovery rate (FDR) < 0.0001. OncomiRs expressed at detectable levels in more than 20% of our populations were considered for statistical analysis. This threshold was chosen after looking at the distributions of the miRNA measures allowing for the best compromise between the number of selected miRNAs and their detection prevalence in the population, thereby ensuring that our findings possess broader applicability.

### 2.4. Statistical Analysis

The data consisted of 911 records, including information about 75 EV oncomiR expressions and exposome factors. The latter consisted of numerical variables: age, BMI, residence PM10 exposure (annual average), C-reactive protein (CRP), homocysteine, coffee consumption (cup/week), alcohol consumption (glass/week), number of previous pregnancies; and categorical variables: education (primary school or less, secondary school, high school, university or more), occupation (employee, unemployed, retired, housewife), residence traffic exposure (mild, moderate, high), lifestyle (sedentary, active, sportive), smoking habit (never, former smoker, current smoker), passive smoker (no, yes), menopausal status (no, yes), use of antidepressants (no, yes), use of oral contraceptives (no, yes).

The strategy of analysis sketched in the following section was applied separately for female and male data. In the first step, multivariate relationships among miRNA expressions and exposure variables were investigated via a multivariate analysis approach using Factor Analysis of Mixed Data (FAMD) methods [25,26]. In this context, exposure variables were considered active variables and miRNA expressions were considered supplementary (or passive) variables. According to FAMD methods, the following strategy of analysis was adopted: (1) the number of relevant principal axes that summarize the active variables was chosen by evaluating the indices of the explained inertia; (2) the relationships among the exposure variables and principal axes were evaluated by a plot of squared correlation coefficients; (3) finally, the relationships among the miRNA expressions and exposure variables were evaluated by the plotting the orthogonal projections of the former ones onto the principal axes (passive projections plot).

The “ggstatsplot” package [27] was used to perform Spearman correlation analysis on miRNA expression and “ggplot2” was used to visualize the results by generating a heatmap.

Multivariate Adaptive Regression Spline (MARS) models [28] were fitted to explore non-linear and non-additive relationships between single miRNA expressions and exposure variables. According to MARS methods, a regression model is built by performing two automatic selection steps. In the former one, called the forward step, additive and interaction effects, also including the non-linear effects of numerical variables, are included in the model according to the reduction of a sum-of-squares residual error. Non-linear effects are represented using linear splines called “hinge functions”. In the latter step, called the backward step, the aim is to improve the generalizability of the model determined at the end of the previous step. To such an end, the optimal number of variables is selected using the Generalized Cross-Validation (GCV) statistic and the model is then pruned according to this choice. It is worth noting that the forward step can be customized. First, by specifying the “degree” option, one can choose the maximum order of interactions: degree = 1 means no interactions (in other terms, only additive effects can be included in the model), degree = 2 means order one interaction and so on. Second, in an analogous fashion, it is possible to avoid the specification of the non-linear effects of numerical variables (linearity constraints).

In the current study, miRNA expressions were considered dependent variables and exposure factors were considered candidate predictors. For each EV oncomiR, several models were fitted according to the order of interactions (degree = 1 and degree = 2) and the presence/absence of the constraints on non-linear effects. According to the bias vs. variance trade-off, it is important to acknowledge that linear relationships between response variables and predictors may oversimplify the complexity of real-world relationships; but, they do so with a gain in interpretability. Therefore, in exploratory analyses, we decided to evaluate models with hinge functions along with models with linear effects because the former could be unstable and too complex to be interpreted, despite reducing the bias.

For each of the fitted models, the goodness of fit was evaluated using R^2^ and generalized R^2^ coefficients. The importance of each predictor (i.e., exposure variables) was evaluated by the nsubset [29]. All of the analyses were performed using the R software release 4.2.3 [30], with the packages FactoMineR [31] and Earth [32], and the Knime Analytic Platform release 4.7.0 [33].

## 3. Results

### 3.1. Characteristics of the Study Population

For a total of 911 subjects, 238 males and 673 females were included in this study. A description of the study participants is reported in Table 1.

Briefly, the study participants had mean ages of 50.9 ± 13.3 years and 51.6 ± 13.4 years in males and females, respectively. The mean BMIs were equal to 33.9 ± 4.7 kg/m^2^ (males) and 33.6 ± 5.7 kg/m^2^ (females). Study subjects were, for a large percentage, employed (males 69.8%; females 55.7%), sedentary (males 62.2%; females 64.4%), and non-current smokers (males 82.4%; females 85.5%). The association between the different continuous predictors was assessed by Spearman’s rank correlation coefficients and is reported in Figure 1A (females) and Figure 1B (males) together with scatter plots with smoothing trend lines for continuous predictors.

Concerning females, no major monotonic associations were observed; although, moderate (r > 0.1) correlations involved age with homocysteine, age and number of pregnancies, BMI and CRP, BMI and number of pregnancies, and CRP and alcohol consumption (Figure 1A). Similarly, in males, moderate positive correlations involved age with the PM10 annual average and alcohol consumption, BMI and CRP, BMI and alcohol consumption, CRP and homocysteine, alcohol and homocysteine, coffee consumption, and alcohol consumption(Figure 1B).

Concerning the FAMD, the plot showing the joint contributions of each variable is reported in Figure 2. In females, the first principal component, Dim1, is primarily related to age, menopausal status, and working status variables. Dim2 is predominantly related to BMI and PCR, with physical activity variables also contributing to a lesser extent. Lastly, the third principal component, Dim3, mainly comprises the PM10 annual average.

Conversely, for males, the first principal component, Dim1, is predominantly related to age and working status variables, with smoking also contributing significantly. Dim2 is mainly related to PCR and coffee consumption, where these variables show opposition in their influence, along with traffic and physical activity variables playing a secondary role. Finally, the third principal component, Dim3, is primarily composed of BMI.

### 3.2. miRNA Selection and Expression Levels in the Study Subjects

Appendix A presents a comprehensive list of miRNAs identified from the OncomiR database that were ranked in the top 50 based on their association with specific cancer types. Appendix A includes the percentage of expression in females and males regarding the SPHERE study along with an indication of their recorded average up-regulation/downregulation in cancer tissues. Spearman’s correlation coefficients were calculated and presented in a correlation matrix heatmap, based on the average correlation values for each column and row, and are displayed in Appendix A for females (Panel A) and males (Panel B).

In Figure 3A, the overlapping miRNAs across various cancer types are displayed. In Panel B, we present the miRNAs observed in female cancers, expressed in at least 20% of the women included in this study. Similarly, Panel C reports the equivalent selection of miRNAs for male cancers.

Additionally, Figure 4 and Figure 5 report (in females and males, respectively) the relationships of the miRNA subsets for each cancer type as supplementary (or passive) variables and continuous predictors (active variables). Only major associations are reported because the small observed correlations between the sets of active and passive variables lead to most of the miRNA projections that tend to gather very close to the centroid of the axes.

### 3.3. Multivariate Adaptive Regression Splines

The MARS algorithm was applied to explore the non-linear and non-additive relationships between miRNA expression and the selected risk factors in order to evaluate the importance of the explanatory variables we considered. Appendix A reports the results of the models we applied in the female group of subjects while the models for the male group are reported in Appendix A. In addition, both of the above tables include additional results obtained by fitting models with continuous predictors constrained to linear effects. Finally, the goodness of fit of each model, evaluated using R2 and generalized R2 coefficients, is described in Appendix A.

In females, as we considered MARS models with Degree 1 (i.e., MARS model using only additive terms to fit the data), the higher R-squared value for the model was observed for hsa-miR-136-5p, indicating that 25% of the variance in this miRNA expression could be explained by the selected risk factors (PM10, BMI, passive smoke, working status, and homocysteine). This association is coherent with the projections of continuous variables from the FAMD analysis of Figure 5 showing the major role of hsa-miR-136-5p as a passive variable. As we considered MARS models with Degree 2 (i.e., MARS model considering also interactions among predictors to fit the data), the higher R-squared value for the model was observed for hsa-miR-301a-3p, indicating that 29% of the variance in this miRNA expression could be explained by a complex combination of selected risk factors (age: alcohol, PM10 with antidepressants, PCR, working status and alcohol, antidepressants with education, PCR, homocysteine, birth control, number of pregnancies, consumption of coffee, PCR with smoking status and alcohol consumption, and alcohol consumption with physical activity.

As we considered MARS models with Degree 1 and hinges, in males, the higher R-squared value for the model was observed for hsa-miR-136-5p, indicating that 37% of the variance in this miRNA expression could be explained by the selected risk factors (PM10 annual average, age, and alcohol). As we considered MARS models with Degree 2, the higher R-squared value for the model was observed for hsa-miR-338-5p, indicating that 61% of the variance in this miRNA expression could be explained by a complex combination of selected risk factors (AgexCoffee; BMIxCoffee, PM10xCoffee; EducationxHomocysteine; PCRxCoffee; HomocysteinexCoffee; SmokingxCoffee; Physical ActivityxCoffee; Working statusxCoffee; and AlcoholxCoffee).

Intriguingly, the PM10 annual average emerged as the top-ranked variable in terms of its importance for numerous miRNAs, both in females (Appendix A) and males (Appendix A).

Given the complexity of the results, we here provide detailed descriptions for the associations involving predictors with a top-ranking priority (i.e., 1st). Individual raw outputs are reported in the Appendix A.

Age was identified as a robust predictor of miRNA expression in females with hsa-miR-324-3p, hsa-miR-671-3p, hsa-miR-7-1-3p, and hsa-miR-9-5p, for which the predictor showed maximum importance. In males, age was similarly found to be a key predictor, correlating with hsa-miR-574-3p, hsa-miR-125a-5p, and hsa-miR-126-3p. Notably, a distinctive link between age and hsa-miR-125a-5p was observed exclusively in males; meanwhile, in females, age ranked as the second most important factor influencing miR-125a-5p expression.

Furthermore, BMI showed notable associations with miRNA expression, specifically hsa-miR-378a-3p and hsa-miR-708-5p in females and hsa-miR-9-5p, hsa-miR-103a-3p, and hsa-miR-140-5p in males. Interestingly, hsa-miR-378a-3p displayed a remarkable association with BMI, ranking fourth among the associated predictors for males.

Regarding biochemical parameters, CRP emerged as the top factor associated with the modulation of hsa-miR-338-5p in both females and males. In addition, homocysteine exhibited significant associations with hsa-miR-590-5p, ranking as the primary predictor in females and the secondary predictor in males.

The consumption of coffee demonstrated a significant correlation with miRNA expression in both females (hsa-miR-144-5p, hsa-miR-130b-3p, and hsa-miR-7-1-3p) and males (hsa-miR-139-3p, hsa-miR-378a-3p, and hsa-miR-148a-3p).

Lastly, hsa-miR-26b-3p exhibited a strong association with physical activity in females (as related to BRCA) but was not included in the male oncomiRNA list we selected a priori; thus, it was not further investigated.

## 4. Discussion

The present study investigated the association between selected oncomiRNAs and various factors, including age, BMI, biochemical parameters, and coffee consumption, shedding light on potential regulatory mechanisms and their relevance in the context of exposome research. Moreover, we investigated the gender-specific patterns observed in the associations, emphasizing the importance of considering this variable in miRNA-related research. Our objective is to enhance the comprehension of the complex association dependence of miRNAs with diverse factors. This highlights the importance of developing multivariable methodological approaches for overcoming the analysis of one exposure at a time, as is conventionally performed in exposome research. Emphasizing the relevance of considering non-linear relationships among variables, we seek to advance the understanding of miRNA regulation in response to multiple environmental influences. By exploring the broader context of miRNA regulation, our findings could offer valuable insights into the complexity of molecular interactions and pave the way for more comprehensive and robust exposome studies in the future.

In our investigation, age emerged as a robust and influential predictor of miRNA expression in both female and male subjects. Specifically, we observed major associations between age and the expression levels of miR-324-3p, miR-671-3p, miR-7-1-3p, and miR-9-5p in females and miR-574-3p, miR-125a-5p, and miR-126-3p in males.

An intriguing finding in our study was the gender-specific association of age with miR-125a-5p. Age significantly influenced miR-125a-5p expression in both females and males, providing robust evidence for the role of this association in two independent sets of subjects. In males, however, age was the single selected predictor of miR-125a-5p while a more complex pattern of modulation was observed in females. Namely, the hinge model showed the role of PM10, age, BMI, and PCR ranked by importance; the same variables (plus coffee consumption) were observed in Degree 2 models with interactions. Under linear constraint age, PM10, menopause, and BMI were showing their ranked importance. Considering linear interactions, the selected models showed the major role of being retired and post-menopause. The difference between linear and hinge models could be related to underlying non-linear and non-additive dependence relations on the considered continuous variables possibly surrogated by the associated categorical factors of menopause and working status.

These results suggest that age plays a central role in shaping miRNA expression profiles in both genders. Furthermore, the gender-specific associations underscore the importance of considering sex as a biological variable in miRNA-related studies as it may uncover unique regulatory patterns and provide insights into gender-specific molecular mechanisms associated with age-related processes.

Additionally, miR-125a-5p has been extensively studied in the context of cancer [34,35]. It acts as a tumor suppressor in various cancers by targeting oncogenes or genes involved in cell proliferation, invasion, and metastasis [36,37]. Its down-regulation in cancer cells can contribute to tumor progression [38,39]. Our findings confirm the age-dependent down-regulation in the expression of miR-125a-5p, which has been previously reported in mouse models [40].

We further observed significant associations between BMI and miRNA expression, with gender-specific patterns present. In females, miR-378a-3p and miR-708-5p showed an association with BMI; meanwhile, in males, miR-9-5p, miR-103a-3p, and miR-140-5p were found to be associated. These results suggest distinct miRNA regulatory patterns in relation to BMI in females and males, reflecting potential gender-specific molecular mechanisms linked to adiposity and metabolic processes. The SPHERE population consisted of individuals with high BMIs and previous studies have emphasized the significant role of BMI in modulating EV-miRNA expression [21]. Specifically, miR-378a has been implicated in adipogenesis and obesity [41,42,43,44] while miR-708-5p was found to be up-regulated in individuals with obesity and metabolic syndrome [45].

In the context of obesity, miR-9-5p has been frequently reported as dysregulated in adipose tissue and serum, pointing to its potential involvement in the pathogenesis of obesity [46]. It is known to influence adipocyte differentiation by targeting key genes in adipogenesis [47]. Moreover, miR-9-5p has been associated with the regulation of inflammatory processes [48,49]; given that obesity is characterized by chronic low-grade inflammation in adipose tissue [50], the dysregulation of miR-9-5p in obesity may contribute to adipose tissue inflammation and insulin resistance. Similarly, miR-103a-3p has been found to be down-regulated in blood samples of individuals with obesity [51,52] and its role in modulating insulin sensitivity is well-established [52]. Additionally, miR-140-5p has been reported to be up-regulated in the plasma of obese or diabetic patients and its levels can be influenced by treatment with metformin or bariatric surgery, indicating a potential correlation with insulin sensitivity [52,53]. Moreover, miR-140-5p overexpression has been positively correlated with BMI and waist-to-hip ratio [53].

In our investigation of biochemical parameters, we identified CRP as the most influential factor regulating miR-338-5p expression in both females and males. Furthermore, homocysteine showed significant associations with miR-590-5p, ranking as the primary predictor in females and the secondary predictor in males. To the best of our knowledge, no previous studies have reported an association between CRP and miR-338-5p. However, a recent paper demonstrated that overexpressed miR-338-5p effectively suppressed IL-6 expression at both the mRNA and protein levels, both in vitro and in vivo, and vice versa [54]. Additionally, luciferase reporter assays confirmed that miR-338-5p directly regulated IL-6 expression by binding to its mRNA 3′ untranslated region [55]. Given that IL-6 is a known inducer of hepatic C-reactive protein (CRP) synthesis, this discovery suggests a potential link between miR-338-5p and CRP [54]. These findings shed light on a novel regulatory mechanism that may have implications for our understanding of the interplay between miRNAs and inflammatory pathways and could provide insights into the molecular basis of miR-338-5p modulation by CRP.

MiR-590-5p plays a crucial role in cellular homeostasis, including cancer, with recent evidence showing its dual function as both an oncogene and a tumor suppressor, depending on the specific cancer context [56]. Although the link between miR-590-5p and homocysteine is not clear, miR-590-5p has been implicated in processes related to vascular endothelial function [57] and inflammation [58], which are pathways closely connected to homocysteine metabolism. However, it is important to note that the relationship between miR-590-5p and homocysteine is likely to be influenced by various factors related to the cellular context, including other molecular interactions. More comprehensive studies are needed to elucidate the specific mechanisms underlying this potential link and its functional consequences in health and disease.

The significant correlation we found between coffee consumption and miRNA expression in both females and males is a fascinating finding. Coffee contains numerous bioactive compounds, such as caffeine and polyphenols, which have been shown to exert protective effects against cancer by acting as antioxidants and modulating various cellular pathways [59]. Numerous epidemiological studies have explored the association between coffee consumption and cancer risk, revealing intriguing findings. Overall, the evidence suggests that moderate coffee consumption is associated with a potentially reduced risk of certain cancers. Some studies have shown an inverse relationship between coffee consumption and the risk of developing cancers, such as liver, colorectal, and endometrial cancer [60]. Moreover, coffee consumption has been linked to a lower risk of certain types of hormone-related cancers, like estrogen-receptor-negative breast cancer [61]. In the literature, we have not found any possible associations between coffee consumption and the expression of the microRNAs we identified (miR-144-5p, miR-130b-3p, miR-7-1-3p, miR-139-3p, miR-378a-3p, and miR-148a-3p). This field of study is still in its very early stages and further research is required to assess the effect of these dietary compounds.

The intriguing link between miR-26b-3p and physical activity sheds light on the complex regulatory network of microRNAs in response to both exercise and everyday physical behaviors. It emphasizes miR-26b-3p’s potential role as a significant player in processes related to physical activity. Notably, in this context, “physical activity” encompasses habitual behaviors beyond structured exercise. However, to fully comprehend the functional significance and implications of miR-26b-3p in relation to physical activity, further comprehensive studies are warranted.

One of the major advantages of a machine learning method like MARS is its ability to automatically select relevant variables from a large set of predictors and their interactions, also considering non-linear relationships under their hinge function representation. This provides an additional level of information with respect to multivariate projection techniques like FAMD and simple univariable linear regression. Whereas the MARS method can capture non-linear and non-additive relationships, the resulting models are still relatively interpretable, providing relevant exploratory information for setting biological hypotheses and building refined models with optimal control of the bias vs. variance tradeoff and enhanced predictive capability. Therefore, we are aware of the limitations as well as the advantages of the applied MARS method; more sophisticated modeling, rather than multivariable regression, is requested by the integrative analysis of multi-omics data in order to infer about regulatory/causal networks. This is outside of the scope of the present paper but lays the ground for future developments of our analyses of the SPHERE data.

## 5. Conclusions

This study not only enhances our understanding of miRNA regulation in response to multiple environmental and lifestyle influences but also underscores the critical role of methodology in unraveling the complexities of cancer risk factors. By uncovering non-linear relationships among variables, we gain valuable insights into the complexity of miRNA regulatory networks. Furthermore, our adoption of the MARS methodology, with its inherent capacity to automatically identify non-linear and non-additive effects (interactions) among predictors, exemplifies the power of computational techniques in elucidating the interplay between multifaceted variables. This innovation approach not only advances our comprehension of EV oncomiR network regulation but also serves as a model for future studies seeking to decipher complex molecular systems. As we move forward, we envisage a horizon of comprehensive exposome studies that will further define the dynamic relationships between our environment, our genetic makeup, and the complex landscape of cancer development.

## Figures and Tables

**Figure 1 cancers-15-04317-f001:**
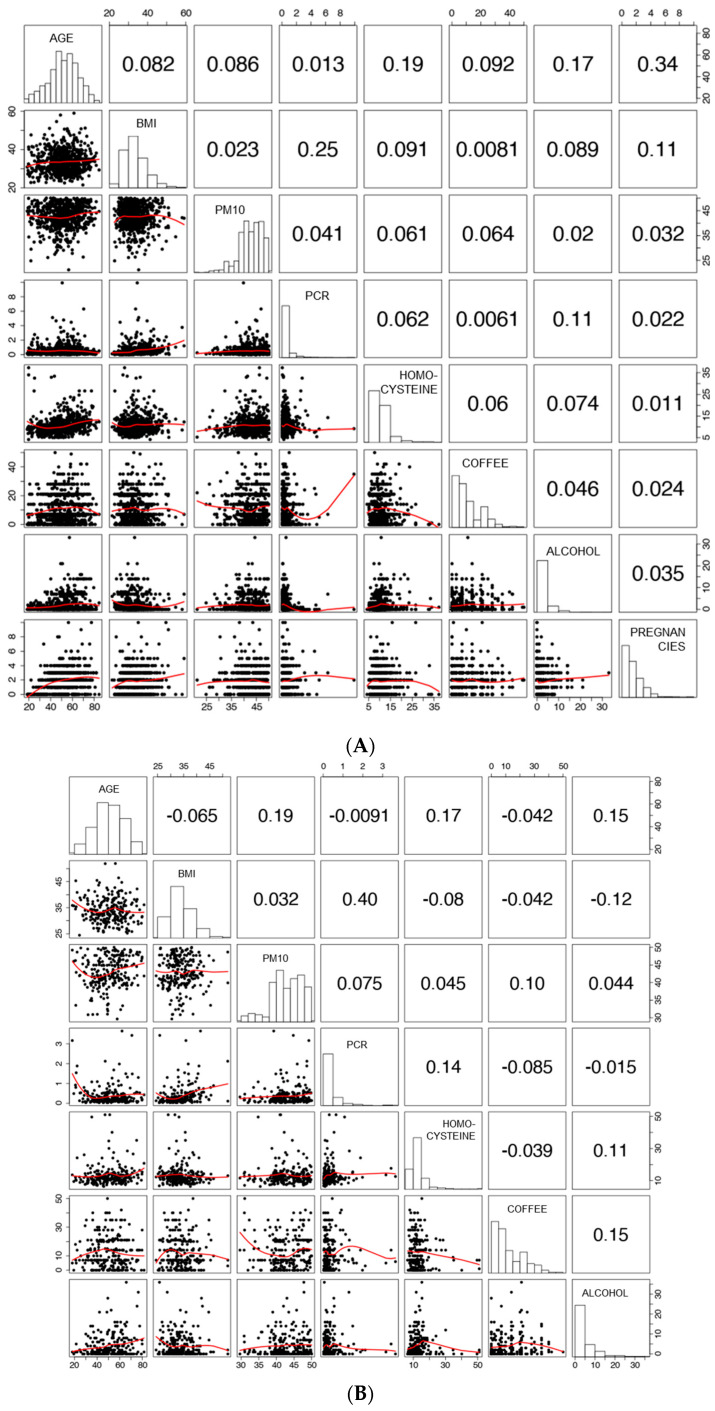
Correlation among numerical variables. (**Panel A**) Female data. (**Panel B**) Male data. On the lower panels, scatterplots for each variable pair are reported. The red curves were obtained by the non-parametric regression method (lowess smoother). On the upper panel are the reported Spearman correlation coefficients for each variable pair.

**Figure 2 cancers-15-04317-f002:**
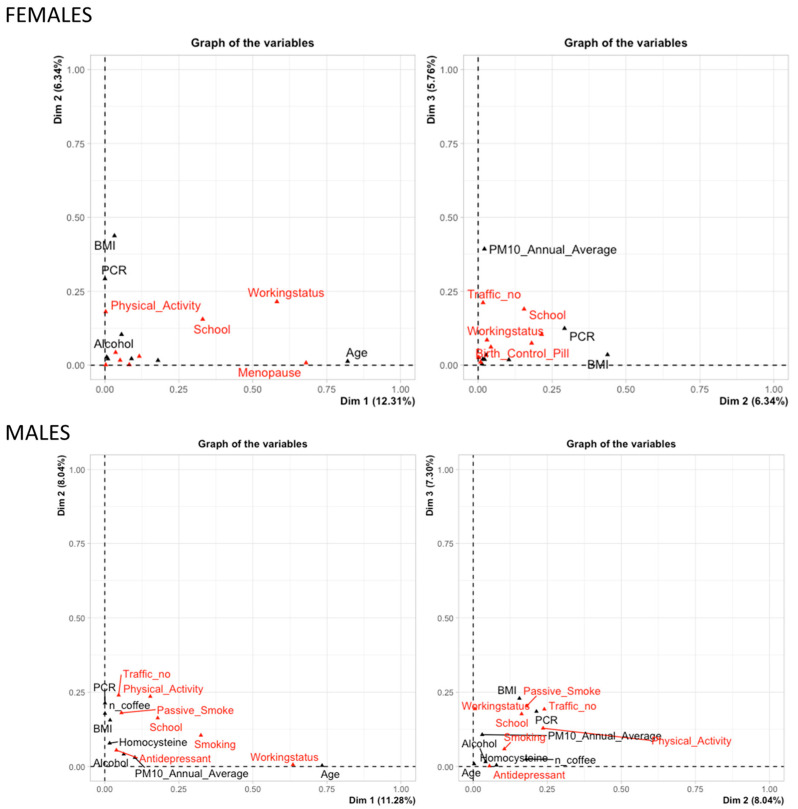
Multivariate analysis of miRNA expressions and exposure variables. Factor Analysis of Mixed Data (FAMD) was used to investigate the joint contributions of various variables in females and males. Continuous variables are reported in red and categorical variables are in black. Labels of variables with minor contributions were omitted.

**Figure 3 cancers-15-04317-f003:**
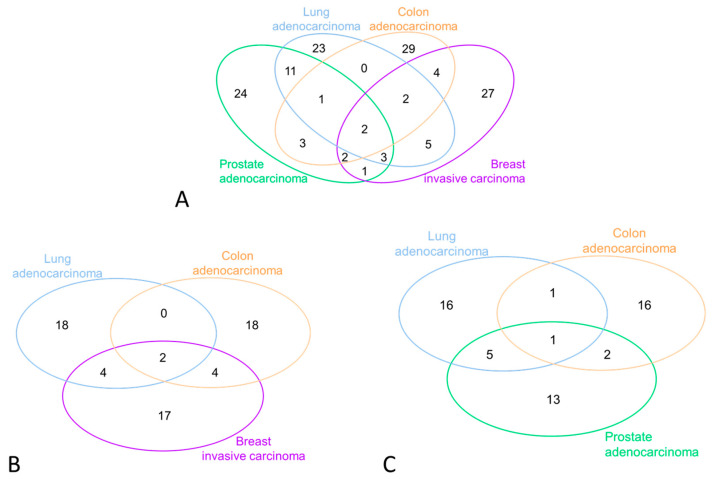
miRNA expression patterns in different cancer types. (**Panel A**) illustrates overlapping miRNAs across various cancer types (females AND males). (**Panel B**) displays miRNAs observed in female cancers, expressed in at least 20% of the women in this study. (**Panel C**) presents the equivalent selection of miRNAs for male cancers.

**Figure 4 cancers-15-04317-f004:**
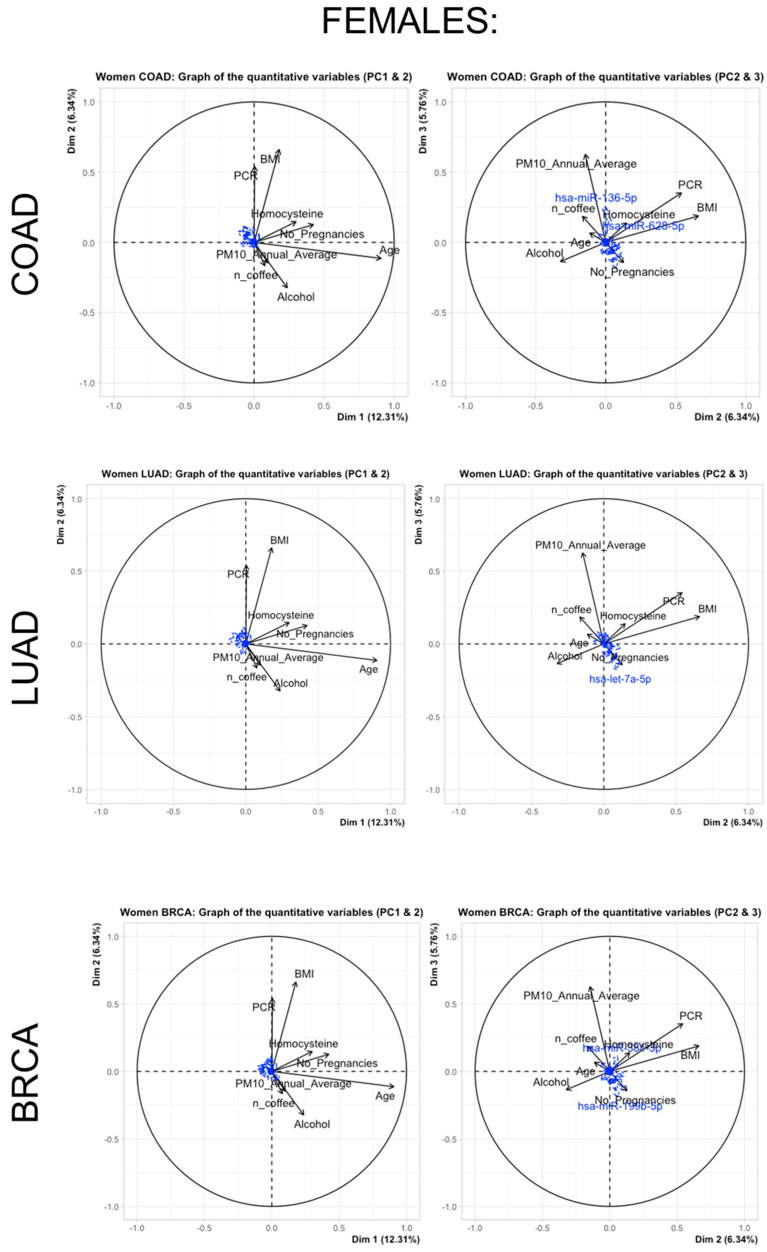
Relationships of miRNA subsets for each cancer type in females (Colon cancer—COAD, Lung cancer—LOAD, Breast cancer—BRCA). The miRNA subsets are presented as supplementary (passive) variables while continuous predictors are represented as active variables.

**Figure 5 cancers-15-04317-f005:**
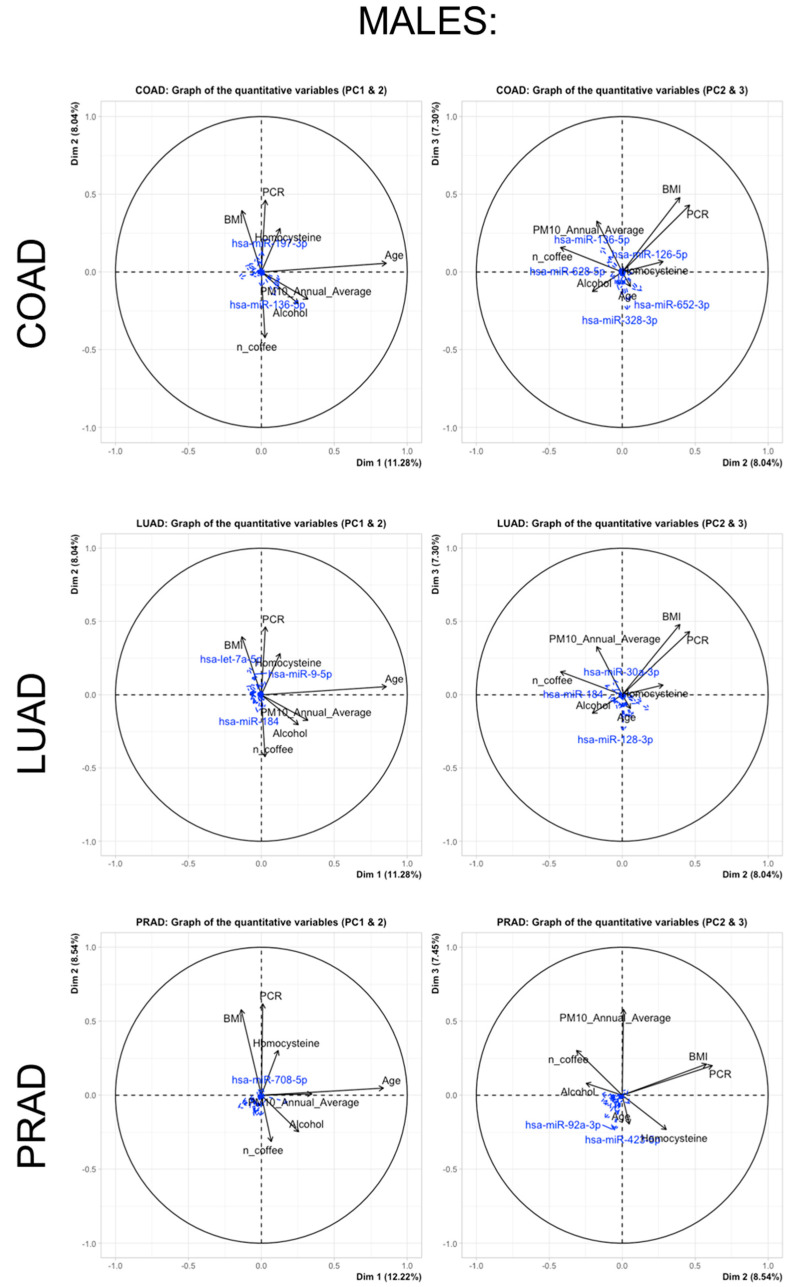
Relationships of miRNA subsets for each cancer type in males (COAD, LOAD, PRAD). The miRNA subsets are presented as supplementary (passive) variables while continuous predictors are represented as active variables.

**Table 1 cancers-15-04317-t001:** Description of the study population.

		Males	Females
		(N = 238)	(N = 673)
Age	years, mean ± SD	50.9 ± 13.3	51.6 ± 13.4
BMI	kg/m^2^, mean ± SD	33.9 ± 4.7	33.6 ± 5.7
Education	Primary school or less	21 (8.8%)	85 (12.6%)
	Secondary school	61 (25.6%)	178 (26.4%)
	High school	117 (49.2%)	315 (46.8%)
	University or more	39 (16.4%)	95 (14.1%)
Occupation	Employee	166 (69.8%)	375 (55.7%)
	Unemployed	15 (6.3%)	55 (8.2%)
	Retired	57 (23.95%)	160 (23.8%)
	Housewife	-	83 (12.3%)
Residence traffic exposure	Mild	53 (22.2%)	148 (22.0%)
	Moderate	117 (49.2%)	329 (48.9%)
	Heavy	68 (28.6%)	196 (29.1%)
PM10 annual average	from the ARPA monitoring station µg/m^3^, mean ± SD	43.0 ± 4.4	42.7 ± 4.7
Physical activity	Sedentary	148 (62.2%)	433 (64.4%)
	Active	66 (27.7%)	206 (30.6%)
	Sportive	24 (10.1%)	34 (5.0%)
Antidepressants	Yes	15 (6.3%)	100 (14.9%)
	No	223 (93.7%)	573 (85.1%)
Smoking	Never	88 (37.0%)	377 (56.0%)
	Ex	108 (45.4%)	198 (29.4%)
	Current	42 (17.6%)	98 (14.5%)
Passive smoke exposure	Yes	110 (46.2%)	293 (43.5%)
	No	128 (53.8%)	380 (56.5%)
Alcohol	weekly consumption, median [Q1, Q3]	2 [0; 7]	0 [0; 2]
Coffee	weekly consumption, median [Q1, Q3]	9 [5; 20]	7 [3; 16]
C-reactive protein	mg/L, median [Q1, Q3]	0.22 [0.11; 0.45]	0.31 [0.14; 0.6]
Homocysteine	µmol/L, median [Q1, Q3]	12 [10.2; 14.2]	9.8 [8.3; 12.1]
Birth control pill	Yes	-	26 (3.9%)
	No		647 (96.1%)
Number of previous pregnancies	median [Q1, Q3]	-	2 [1; 3]
Menopause	Yes	-	355 (55.1%)
	No		287 (44.9%)

## Data Availability

The data presented in this study are available on request from the corresponding author.

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
