# Peer review of "Environmental and Lifestyle Cancer Risk Factors: Shaping Extracellular Vesicle OncomiRs and Paving the Path to Cancer Development"

_cancers, 2023, doi:10.3390/cancers15174317_

Round 1

Reviewer 1 Report

The study focuses on EV derived miRNA profile in regard to cancer incidence. Thus the topic seems rather attractive but its importance is lost due to complicated comparisons of too many parameters. 

It is based mostly on bioinformatics and evaluation of metadata. The data obtained by miRNA profiling were not verified by other, more specific method. 

However, the study aimed for a complex insight on the EV / miRNA /lifestyle relationship and the aim was accomplished. Considering the amount of comparisons and factors, the text is written very well and explanatory.

Author Response

Comment

The study focuses on EV derived miRNA profile in regard to cancer incidence. Thus the topic seems rather attractive but its importance is lost due to complicated comparisons of too many parameters.

It is based mostly on bioinformatics and evaluation of metadata. The data obtained by miRNA profiling were not verified by other, more specific method.

However, the study aimed for a complex insight on the EV / miRNA /lifestyle relationship and the aim was accomplished. Considering the amount of comparisons and factors, the text is written very well and explanatory.

Reply

We appreciate the reviewer's comments on our study focusing on EV-derived miRNA profiles in relation to cancer risk factors. We value the opportunity to address the concerns raised and provide further clarification on the points highlighted.

The reviewer observed that our study involves a comprehensive examination of multiple features. We acknowledge that this approach can present challenges in terms of clarity and focus. Given that cancer development is a multifaceted process influenced by a multitude of factors, the concept of risk assessment inherently necessitates the consideration of numerous parameters. Our aim was indeed to capture the intricate interplay of these factors and their potential impact on miRNA profiles within EVs.

The reviewer rightfully highlights our study's utilization of bioinformatics and metadata analysis. It is important to emphasize that while these methods were instrumental in the initial selection of candidate miRNAs for our investigation, the core measurement of miRNA expression was carried out using real-time quantitative polymerase chain reaction (qRT-PCR). We recognize that qRT-PCR is widely regarded as the gold standard for miRNA quantification due to its high analytical sensitivity and specificity. Given the established reliability of qRT-PCR, the data obtained from this method inherently reflect the validated expression levels of the selected miRNAs. In light of this, we concur that the miRNA expression data itself, acquired through qRT-PCR, does not necessitate further validation as it is already based on a well-established and rigorously validated methodology. The role of bioinformatics and metadata analysis in our study was primarily to guide the selection of miRNAs of interest, enhancing the efficiency of our investigation. The bioinformatics approach facilitated the identification of miRNAs that demonstrated potential relevance to our research questions, allowing us to focus our subsequent qRT-PCR experiments on these candidates.

We are pleased that the reviewer recognizes the accomplishment of our study's overarching goal—to provide a comprehensive understanding of the intricate relationship between EVs, miRNAs, and lifestyle factors in the context of cancer development. Our intention was in fact to offer a integrated view of this complex interplay, and we are grateful that our efforts have been considered successful in this regard.

Reviewer 2 Report

In their ntitled manuscript “Environmental and Lifestyle Cancer Risk Factors: Shaping Ex-2 tracellular Vesicle-oncomiRs and Paving the Path to Cancer De-3 velopment”, the authors present a compelling investigation into the plasma levels of EV-borne oncomiRs within a specific population. They skillfully analyze the top oncomiRs associated with prevalent cancers in both women and men, selecting only those expressed in more than 20% of the population for their statistical study. Through the use of a multivariate adaptive regression splines (MARS) model, they explore complex interactions between environmental and lifestyle factors and EV-oncomiRs. Their innovative approach provides valuable insights into the nonlinear relationships among these variables and enhances our understanding of miRNA regulatory networks. This work is significant and promising, paving the way for future comprehensive exposome studies. It offers a fresh perspective on how we might approach cancer risk factors and is certainly worth reading for those interested in this field. The work is interesting, but there are several comments that should be addressed before considering publication.

1. In this work, only oncomiRs expressed in more than 20% of the population were considered for statistical analysis, a decision guided by specific criteria that would need further explanation regarding the logic behind this threshold choice.

2. The raw sequence data should be uploaded to a public database to ensure transparency and facilitate further research by other scientists in the field. This work is an important resource paper; without the open raw data, its meaningfulness is limited.

3. The font size and type of the figures should remain consistent. Please ensure that the font type used is either Times New Roman or Arial.

The lanuage should be improved.

Author Response

Comment

In their entitled manuscript “Environmental and Lifestyle Cancer Risk Factors: Shaping Extracellular Vesicle-oncomiRs and Paving the Path to Cancer Development”, the authors present a compelling investigation into the plasma levels of EV-borne oncomiRs within a specific population. They skillfully analyze the top oncomiRs associated with prevalent cancers in both women and men, selecting only those expressed in more than 20% of the population for their statistical study. Through the use of a multivariate adaptive regression splines (MARS) model, they explore complex interactions between environmental and lifestyle factors and EV-oncomiRs. Their innovative approach provides valuable insights into the nonlinear relationships among these variables and enhances our understanding of miRNA regulatory networks. This work is significant and promising, paving the way for future comprehensive exposome studies. It offers a fresh perspective on how we might approach cancer risk factors and is certainly worth reading for those interested in this field. The work is interesting, but there are several comments that should be addressed before considering publication.

  1. In this work, only oncomiRs expressed in more than 20% of the population were considered for statistical analysis, a decision guided by specific criteria that would need further explanation regarding the logic behind this threshold choice.
  2. The raw sequence data should be uploaded to a public database to ensure transparency and facilitate further research by other scientists in the field. This work is an important resource paper; without the open raw data, its meaningfulness is limited.
  3. The font size and type of the figures should remain consistent. Please ensure that the font type used is either Times New Roman or Arial.

Reply

We appreciate the reviewer's thoughtful assessment of our manuscript. We are pleased to know that the investigation into EV-borne oncomiRs and their interactions with environmental and lifestyle factors has been acknowledged as compelling and innovative.

  1. We thank the reviewer for highlighting the criterion we applied in selecting oncomiRs for statistical analysis, specifically those expressed in more than 20% of the population. This decision was made based on the aim to focus our analysis on miRNAs with a relatively prevalent presence in the population, in order to provide insights that could be more broadly applicable. Moreover, applying statistical modelling to data with truncated distributions, adds an additional layer of complexity we avoid in the current paper but we planned to face in further investigations, to enhance the extraction of relevant information from these data. We have now included a dedicated subsection in the manuscript's methodology section, outlining the logic and considerations that guided our decision.

“In total, 545 miRNAs were included in the analysis after the exclusion of non-amplified miRNAs. The application of the global mean was identified as the optimal normalization method, and miRNA expression levels were quantified using relative quantification, represented as 2^-ΔCt. Leveraging the OncomiR database [23], we extracted the top 50 oncomiRs for each of the three most prevalent cancers in both women (breast, colorectal, and lung carcinomas) and men (lung, prostate, and colorectal carcinomas) [24]. Notably, this selection process was underpinned by stringent criteria, including a significance threshold of p-value < 0.0001 and false discovery rate (FDR) < 0.0001. OncomiRs expressed at detectable levels in more than 20% of our populations were considered for statistical analysis. This threshold was chosen after looking to the distributions of miRNAs measures allowing the best compromise between the number of selected miRNAs and their detection prevalence in the population, thereby ensuring that our findings possess broader applicability”.

  1. We fully acknowledge the importance of transparency and the accessibility of research data. To ensure reproducibility and further exploration of our findings by other researchers, we will make the raw data available upon request. Interested parties can contact us for access to the data, thereby facilitating future research endeavors in this area, according to the policy of the funded project.
  2. We appreciate the reviewer's attention to detail regarding the presentation of figures. We apologize for any early inconsistencies in font size and type within the figures. We will conduct a thorough review of all figures to ensure that font size and type remain consistent throughout the manuscript (i.e. Arial).

Reviewer 3 Report

NICE WORK,CONGRATS!

IN INTRODUCTION I WOULD PREFER A LESS DETAILED AND BETTER STRUCTURED TEXT,JUST TO GIVE THE MAJOR POINTS OF THE ARTICLE.MATERIALS,RESULTS AND DISCUSSION JUST FINE. THE CONCLUSION PART IS RATHER POOR,YOU SHOULD BE MORE EXPLAINATORY EITHER IN WHAT YOU HAVE FOUND OR THE FUTURE RESEARCH,BASED ON THE PRESENT WORK.

Author Response

Comment

NICE WORK, CONGRATS!
In introduction I would prefer a less detailed and better structured text, just to give the major points of the article. materials, results and discussion just fine. The conclusion part is rather poor, you should be more explanatory either in what you have found or the future research, based on the present work.

Reply

We sincerely appreciate your kind words and congratulations on our work. We also appreciate your valuable suggestions for enhancing the structure and content of the manuscript.

We acknowledge your recommendation for a more concise and well-structured introduction that highlights the major points of the article. We revised the introduction as follows:

“Environmental and lifestyle factors exert a profound influence on chronic diseases, particularly in the context of cancer development and related clinical outcomes [1,2]. These factors encompass diverse elements [3,4], ranging from exposure to environmental carcinogens and dietary habits [5], to physical activity levels [6], adiposity [7], smoking, alcohol consumption [8], and socioeconomic status [9]. The intricate interplay between these factors and cancer etiology underscores the necessity for a comprehensive approach to cancer prevention and management. By addressing the environmental and lifestyle determinants of cancer, we can implement targeted interventions and promote healthier behaviors [10], aiming to reduce non-intrinsic cancer risk and enhance clinical outcomes in a personalized way.

Individuals with pre-existing hypersusceptibility conditions, such as those associated with adiposity (excessive body fat), might experience an amplified influence from the exposome, i.e., an accumulation of lifelong environmental exposures [11,12]. Adiposity-related conditions, like overweight/obesity and metabolic syndrome, foster a hormonally dysregulated and pro-inflammatory environment both at systemic and tissue levels. This inflamed host environment can synergize with external exposures, augmenting the vulnerability to cancer development and progression [11,13].

Extracellular vesicles (EVs), encompassing exosomes and microvesicles, are small membrane-bound structures released by various cell types [14]. Once considered cellular debris, EVs are now recognized as sophisticated carriers of cellular cargo, including proteins, lipids, and nucleic acids [15]. Owing to their ability to traverse bio-logical barriers, EVs serve as conduits for intercellular communication and facilitate information exchange between cells, even across distant locations within the body [14]. These distinctive attributes render EVs versatile entities with significant roles in diverse physiological processes and disease contexts, including cancer [16,17].

MicroRNAs (miRNAs) are pivotal regulators of gene expression among the myriad of molecules transported by EVs. MiRNAs, short non-coding RNA molecules, play a critical role in post-transcriptional gene regulation by targeting messenger RNAs (mRNAs) for degradation or translational inhibition [18]. Within this group, oncomiRs, a subset of miRNAs, have garnered attention for their involvement in cancer-related processes, including tumor initiation, growth, metastasis, and therapy resistance [19,20].

Understanding the mechanisms driving the selective packaging of oncomiRs into EVs and their subsequent release into the extracellular space has become central in cancer research. EVs enriched with oncomiRs can engage in cell-to-cell communication, delivering functional oncomiRs to recipient cells—ranging from neighboring cancer cells to fibroblasts, adipocytes, and immune cells. This transfer reshapes the gene expression profiles and behavior of recipient cells, contributing to the ability of cancer cells to shape both local and distant microenvironments. EVs containing miRNAs thus dynamically orchestrate multidirectional communication, serving as the hubs of the intercellular network, ultimately transforming the microenvironment in a pro-tumoral manner. This intriguing phenomenon fuels interest in cancer biology, as it may hold substantial implications for innovative diagnostics and therapeutic strategies.

The SPHERE project [21], with its unique attributes, has enabled the investigation of EVs and their miRNA content's role in mediating the impact of air particulate matter (PM) on cardiovascular health. SPHERE offers comprehensive insights into the microRNA content of EVs within a population of over 2000 individuals characterized by a body mass index (BMI) exceeding 25 and a well-defined exposome.

In this paper, we delve into the captivating domain of EVs and oncomiRs, probing the oncomiRs associated with specific cancer types based on their dependency on major exposome factors in the SPHERE screening subpopulation of 911 non-oncological subjects with assessed miRNome.

Given the complexity of analyzing these data, it's essential to consider the application of inductive inference methods rooted in machine learning, extending multivariate analysis and regression approaches. In this context, adopting automated methods for variable selection and their preferably non-linear/non-additive effects on EV-contained miRNA expression is pivotal. Within the realm of multiple regression models, the multivariate adaptive regression splines (MARS) methodology appears promising, given its ability to automatically identify non-linear effects and interactions (effect modifiers) among predictors.”

Thank you for highlighting the importance of a robust conclusion that either elaborates on our findings or outlines potential avenues for future research. We revised the conclusion to provide a more comprehensive summary of our key findings, while also offering insights into potential future directions that arise from our current investigation. The revised version is as follows:

“This study not only enhances our understanding of miRNA regulation in response to multiple environmental and lifestyle influences but also underscores the critical role of methodology in unraveling the complexities of cancer risk factors. By uncovering non-linear relationships among variables, we gain valuable insights into the complexity of miRNA regulatory networks. Furthermore, our adoption of the MARS methodology, with its inherent capacity to automatically identify non-linear and non-additive effects (interactions) among predictors, exemplifies the power of computational techniques in elucidating the interplay between multifaceted variables. This innovation approach not only advances our comprehension of the EV-oncomiR network regulation but also serves as a model for future studies seeking to decipher complex molecular systems. As we move forward, we envisage a horizon of comprehensive exposome studies that will further define the dynamic relationships between our environment, our genetic makeup, and the complex landscape of cancer development.”

Round 2

Reviewer 2 Report

Thank you to the author for addressing all of my concerns and comments. The work is now suitable for the journal.